# Metal Oxide-Based Photocatalytic Paper: A Green Alternative for Environmental Remediation

Daniela Nunes *[iD], Ana Pimentel, Rita Branquinho [iD], Elvira Fortunato and Rodrigo Martins *[iD]

Department of Materials Science, NOVA School of Science and Technology, Campus de Caparica, i3N/CENIMAT, 2829-516 Caparica, Portugal; acgp@campus.fct.unl.pt (A.P.); ritasba@fct.unl.pt (R.B.); emf@fct.unl.pt (E.F.)
* Correspondence: daniela.gomes@fct.unl.pt (D.N.); rm@uninova.pt (R.M.);
  Tel.: +351-212948562 (D.N. & R.M.); Fax: +351-21-294-8558 (D.N. & R.M.)

**Abstract:** The interest in advanced photocatalytic technologies with metal oxide-based nanomaterials has been growing exponentially over the years due to their green and sustainable characteristics. Photocatalysis has been employed in several applications ranging from the degradation of pollutants to water splitting, $CO_2$ and $N_2$ reductions, and microorganism inactivation. However, to maintain its eco-friendly aspect, new solutions must be identified to ensure sustainability. One alternative is creating an enhanced photocatalytic paper by introducing cellulose-based materials to the process. Paper can participate as a substrate for the metal oxides, but it can also form composites or membranes, and it adds a valuable contribution as it is environmentally friendly, low-cost, flexible, recyclable, lightweight, and earth abundant. In term of photocatalysts, the use of metal oxides is widely spread, mostly since these materials display enhanced photocatalytic activities, allied to their chemical stability, non-toxicity, and earth abundance, despite being inexpensive and compatible with low-cost wet-chemical synthesis routes. This manuscript extensively reviews the recent developments of using photocatalytic papers with nanostructured metal oxides for environmental remediation. It focuses on titanium dioxide ($TiO_2$) and zinc oxide (ZnO) in the form of nanostructures or thin films. It discusses the main characteristics of metal oxides and correlates them to their photocatalytic activity. The role of cellulose-based materials on the systems' photocatalytic performance is extensively discussed, and the future perspective for photocatalytic papers is highlighted.

**Keywords:** cellulose-based materials; metal oxide nanostructures; photocatalysis; green and sustainable; environmental remediation





## 1. Introduction

This review explores the recent progress concerning the applications of cellulose-based materials in the field of photocatalysis. The main structural characteristics of cellulose and metal oxides as nanostructures or thin films are described. $TiO_2$ and ZnO are the selected metal oxide photocatalysts. The concept of photocatalytic paper is reviewed, and several studies are presented and discussed.

### 1.1. Cellulose-Based Materials

Cellulose is the most abundant biopolymer on earth, and presents unique characteristics including its biocompatibility and recyclability; it is also lightweight, flexible, foldable, biodegradable, and low cost [1]. This material can be extracted from cotton, wood, hemp, rice, algae, bacteria, and several other sources [1,2]. It is widely employed for producing paper, plastics, textile fabrics, among others.

Cellulose consists of repeating glucose monomers, where the polysaccharide with a molecular structure of $(C_6H_{10}O_5)_n$ is linked together through β-1,4-glycosidic bonds by a condensation reaction [1,3]. Each repeating unit contains hydroxyl groups, and their ability to make hydrogen bonds between cellulose chains determines the final cellulose properties and contributes to its high tensile strength [1,4,5]. The formation of fibrils is

resultant from the Van der Waals and intermolecular hydrogen bonds between hydroxyl groups and oxygens of closer molecules that assist the aggregation of several cellulose chains during biosynthesis, resulting, thus, in the fibrils. These fibrils will aggregate and originate microfibrils (5–50 nm in diameter and several microns in length) [3–5]. The overall microfibril structure can comprehend heterogeneous areas, with areas where the cellulose chains are arranged with high crystalline order and other areas with low order (amorphous) [1,3].

Six cellulose crystalline polymorphs are currently known, namely cellulose I, II, III$_I$, III$_{II}$, IV$_I$, and IV$_{II}$, in which cellulose I and II are found in nature [4]. Cellulose II is known to be the most thermodynamicallystable structure, with cellulose I (metastable) converting to II or III [3]. Each polymorph of crystalline cellulose has been systematically investigated and their structure and characteristics are well documented [3,5], so this will not be further covered in this work.

Traditional paper, with cellulose fibers of about 20 µm in diameter, is usually rough (roughness of hundreds of micrometers) [1]. However, cellulose can reach the nanometer size by means of a disintegration process of the micrometer-sized cellulose fibers. Some examples are microfibrillated cellulose (MFC), nanofibrillated cellulose (NFC), and nanocrystalline cellulose (NCC), which are produced with different disintegration processes [1,6,7]. The transparent NCC membrane (Figure 1) is composed of well-organized three-dimensional structures, with long entangled filaments of a cellulose microfibril network. The average width of the cellulose fibers is within 20–50 nm [7]. Another type of nanocellulose is bacterial nanocellulose (BNC). The bacterium *Gluconacetobacter* is normally responsible for the production of BNC membranes (Figure 1), and such a nanomaterial presents enhanced chemical purity, high crystallinity, and low surface roughness and porosity [8]. As expected, the purified cellulose materials have improved intrinsic properties, including a higher Young's modulus and lower thermal expansion coefficient, among others [1]. The purified BNC membranes are also transparent, which allows their integration in several applications that require this specification.

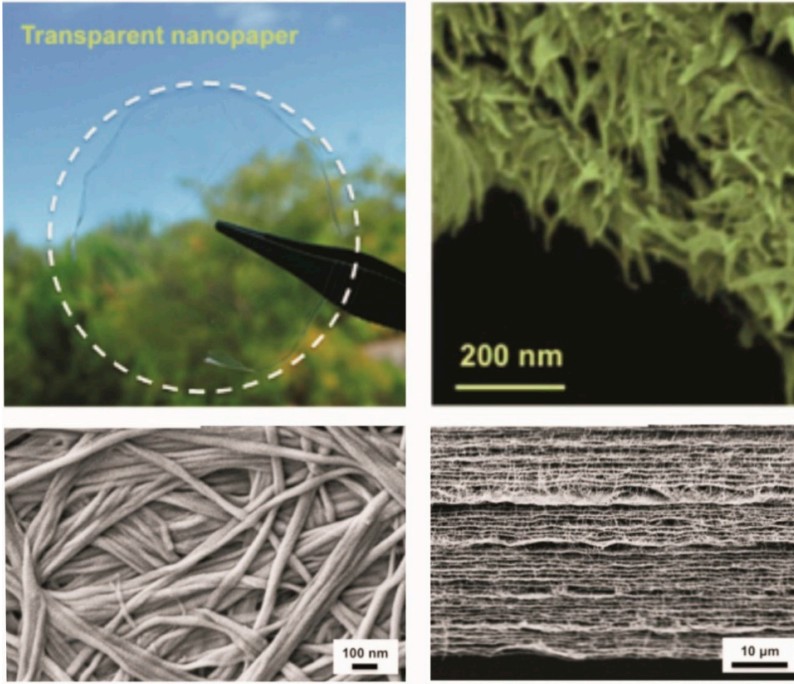

**Figure 1.** Photograph of nanocrystalline cellulose membrane and corresponding scanning electron microscopy (SEM) micrograph. The SEM image of the bacterial cellulose surface (bottom images) obtained from the direct drying process and corresponding cross-section. Reproduced with permission of Elsevier [7].

Cellulose-based materials have been employed in distinct applications, including opto-electronic devices, such as solar cells and transistors [7,9,10], sensors [11–14], electrochromic devices [15,16], but also in diagnostic platforms [17,18] and photocatalysis [12,19,20]. In terms of photocatalysis, despite all the envisioned applications of paper-based materials, its implementation is not straightforward.

### 1.2. Photocatalytic Paper

Extensive efforts have been made by the scientific community and developed countries to promote renewable or carbon-neutral energy sources and find solutions for diminishing the ever-growing industrial residues and human contaminants. Photocatalysis can be considered an appealing strategy to diminish these problems, especially in terms of water purification, with major advantages over the most common methods used, including room temperature reactions, low secondary pollutants' generation and minimal operation costs [21]. Moreover, the photocatalytic process fits in the Sustainable Development Goals by the UN for research and innovation, especially Goal 6, focused in Clean Water and Sanitation [22]. This ever-growing investigation field is reported to have started in 1972, with Fujishima and Honda [23] demonstrating the use of $TiO_2$ for the photoelectrochemical splitting of water using UV light, and further photogeneration of charge carriers in $TiO_2$ [24].

Photocatalysis can be understood as the change of a chemical reaction's rate caused by a photocatalyst (metal oxide material) in the presence of a light source, i.e., ultraviolet (UV), visible (VIS), mixture of both UV+VIS, and solar radiation. There are two types of photocatalytic reactions, i.e., homogeneous and heterogeneous photocatalysis. In homogeneous catalysis, the reactant and catalyst are in the same phase, while in heterogeneous catalysis, both reaction participants are in distinct phases [25]. This process involves excitation, diffusion, and photo-induced charge carriers. A hole-trapping mechanism based on the adsorption/desorption of chemisorbed oxygen molecules at the semiconductor surface governs the photocatalytic process. When the semiconductor is exposed to a light source higher than its band gap ($\lambda < 388$ nm for $TiO_2$ [26] and ~370 nm for ZnO [27]), electron–hole pairs are generated (electrons are generated ($e_{CB}^-$) in the conduction band, while holes ($h_{VB}^+$), in the valence band), in which these photo-generated holes and electrons diffuse to the surface, oxidizing and reducing oxygen and water molecules. With that, reactive radicals ($O_2^{\bullet-}$ and $OH^\bullet$) are created (Figure 2), and such radicals will decompose organic and inorganic compounds on the semiconductor's surface (Figure 2) [25,28,29]. The photocatalytic process will mineralize the organic pollutants into carbon dioxide ($CO_2$), water ($H_2O$) and mineral acids in the presence of the semiconductor photocatalyst and reactive oxidizing species [30]. In brief, the relevant reactions of the organic compounds' photodegradation in the presence of a metal oxide semiconductor are summarized as follows [30–32]:

$$\text{Semicondutor} + h\nu \rightarrow e_{CB}^- + h_{VB}^+ \tag{1}$$

$$H_2O + h_{VB}^+ \rightarrow H^+ + OH^\bullet \tag{2}$$

$$O_2 + e_{CB}^- \rightarrow O_2^{\bullet-} \tag{3}$$

$$O_2^{\bullet-} + H^+ \rightarrow HO_2^\bullet \tag{4}$$

$$HO_2^\bullet + HO_2^\bullet \rightarrow H_2O_2 + O_2 \tag{5}$$

$$H_2O_2 + e_{CB}^- \rightarrow OH^- + OH^\bullet \tag{6}$$

$$H_2O_2 + O_2^{\bullet-} \rightarrow OH^\bullet + OH^- + O_2 \tag{7}$$

$$H_2O_2 + h\nu \rightarrow 2OH^\bullet \tag{8}$$

$$\text{Organic compounds} + OH^\bullet \rightarrow \text{Intermediates} \rightarrow CO_2 + H_2O \tag{9}$$

Photocatalysis has been reported to decompose organic compounds, such as alcohols, carboxylic acids, phenolic derivatives, and aromatic amines and halides, among others [33].

Nevertheless, most photocatalytic studies focus on the degradation of organic pollutants using organic model dyes that simulate pollutants released by industries, textile, pulp and paper, and several other industrial activities, directly to the environment. These organic dyes are complex molecules used to color textiles or products. They are water-soluble, with some being toxic and causing severe impacts on human health [34]. Moreover, these contaminants have a catastrophic effect on the environment and animals, interfering with the reoxygenation capacity of water bodies and blocking sunlight that will directly influence photosynthesis processes [35]. Rhodamine B, methylene blue, methyl orange, congo red, and other azo dyes are examples of model dyes tested in photocatalysis studies [19,35–39]. In terms of inorganic species, bromate, chlorate, halide ions, nitric oxide, palladium, rhodium and sulfur species can be decomposed by photocatalysis, but also metal salts and organometallic compounds [33].

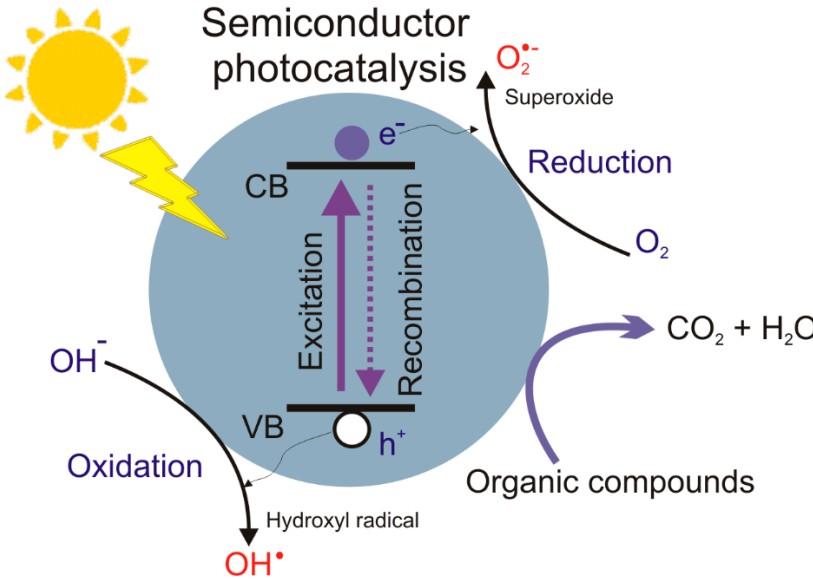

**Figure 2.** Mechanism of semiconductor photocatalysis. Adapted from [40].

The photocatalyst plays a key role in the whole process. The use of nanoparticles in photocatalysis has been largely reported [41–47]; however, the use of nano-photocatalysts in the powder form presents some limitations, especially when it comes to their recovery. The most commonly used commercial $TiO_2$ photocatalyst is also included, with sizes of ~30 nm [48]. Moreover, during the repeated photocatalytic processes, the loss of photocatalysts is highly expected [49]. One way to overcome these limitations is to use membranes, thin films or nanostructures grown/deposited on substrates. In fact, the substrate used, it can contribute to the final photocatalytic activity of the system. Several studies reported photocatalysts grown or deposited on rigid substrates [38,50–52]; however, nowadays the search for flexible substrates is tremendous as these substrates can easily adapt to curved/rough surfaces and decrease production costs. Nanostructures grown on polyethylene terephthalate (PET) [37] or polyethylene naphthalate (PEN) [36], and cotton fabric substrates [53], among others, have been previously reported for photocatalytic applications. Photocatalytic papers have also been previously reported [20,54–56], in which the greater advantage of using paper as a substrate is related to its porous structure that can assist the photodegradation target to diffuse in and the reactive oxygen to migrate out [57]. Moreover, taking advantage of the pore structure created by a reticular fiber network, it has been previously reported that paper was employed as an enhanced support material for gas phase catalytic reactions [55]. Photocatalytic paper is not considered a disposable product anymore, as it can be reutilized several times for the degradation of organic modeling dyes [58].

The main research efforts in photocatalysis are related to materials design and the development of photoreactive systems. The former includes the integration of cellulose-based materials to the photocatalytic system and, in the case of the metal oxide photocatalysts, the search for enhanced nanostructures or thin films for increasing photocatalytic performance.

### 1.3. Metal Oxide Nanostructures and Thin Films

Metal oxides are a class of materials that have been largely investigated over the years due to their unique properties. These materials are known as being highly stable, in some cases, biocompatible, low cost and environmentally friendly, despite their exceptional ability to generate charge carriers under light irradiation [59] with enhanced electrical and optical characteristics and high optical transparency [11,37,38,60–64]. Moreover, at the nanoscale level, these materials can present high surface-to-volume ratios, high surface reaction activity, high catalytic efficiency, and strong adsorption ability [65].

Metal oxides can exhibit different electronic properties, from conducting to semiconducting but also insulating [66]. These materials are known to have the s-shells of positive metallic ions always fully filled by electrons, but the d-shells may not be totally filled [13,67]. This is responsible for the distinct properties of this source of materials, including the wide band gaps that dictate the final photocatalytic behavior of the system. In fact, metal oxides with wide band gaps are active under UV light. However, the solar UV radiation that reaches the earth's surface is relatively small (3–5%) [68], making the use of such materials extremely limited for pollutant degradation with UV radiation. Consequently, solutions to make these materials photoactive under the complete solar spectrum, such as band gap engineering, doping or surface modification, are highly desired [69].

These materials can be divided in two groups: non-transition and transition metal oxides. The non-transition ones include the pre- and post-transition metal oxides. The small energy difference between a cation $d^n$ and either a $d^{n+1}$ or $d^{n-1}$ configuration is responsible for the differences in the material's behavior. Metal oxides with $d^0$ and $d^{10}$ electronic configurations present stable properties, while the pre-transition-metal oxides can be inert in several applications due to their large band gaps, so electrons and holes are hardly formed [13,70]. $TiO_2$, $WO_3$ and $V_2O_5$ are in the transition-metal oxide class of materials with $d_0$ configuration, and $ZnO$ and $SnO_2$ are post-transition metal oxides with $d_{10}$ configuration [70].

Several studies reported the use of metal oxide materials in the photocatalytic degradation of organic contaminants/dyes [19,34,37,38,71–74]. However, their photocatalytic performance strongly depends on the crystal structure, morphology, elemental composition, crystalline phase, exposed surface facets, intrinsic defects (such as oxygen vacancies), lattice distortions, and doping, among others [75–77]. In fact, the presence of oxygen vacancies directly influences the final photocatalytic performance of such materials [78]. Thus, it is of great importance to design and control crystallinity, shape and size, which will directly influence the charge carriers' separation and final photocatalytic activity [24]. Moreover, it is largely known that materials' properties are enhanced at the nanoscale level when compared to bulk counterparts [79]. For that reason, the surface-to-volume ratio of metal oxides is highly increased at the nanoscale, and more active sites on their surface are available for organic molecules to interact, leading to higher overall adsorption capacity and enhanced photocatalytic performance in environmental remediation [35,74].

Zero-dimension (0D) nanomaterials are known to have high specific surface area, while one-dimensional (1D) nanomaterials may result in less recombination due to the short distance for charge carrier diffusion. Two-dimensional (2D) nanomaterials display large surface-to-volume ratios and materials with three-dimension (3D) have higher carrier mobility and abundant active sites, a consequence of porosity and the channels forming its structure [80,81].

Several techniques have been reported to produce metal oxide nanostructures or thin films, including chemical bath deposition [82,83], the sol–gel method [84], electrospinning [83,85], electrodeposition [86,87], magnetron sputtering [88–90], laser assisted

flow deposition (LAFD) [91,92], spray pyrolysis [93,94], and hydrothermal/solvothermal synthesis, either by conventional or by microwave assisted heating [36,62,95–99], and several other methods have been employed. With such a variety of production methods reported, several different nanostructures were described, including nanowires, nanorods, nanofibers, nanotubes, nanobelts, nanowhiskers, nanospheres, tetrapods, nanosheets, and many others [13,71,100–105]. The 1D nanostructures, which include the nanofibers, nanotubes, nanobelts, nanowhiskers, nanorods and nanowires, are extensively investigated for photocatalysis due to their slower electron–hole recombination resulting in higher photocatalytic oxidation rates. Moreover, these 1D materials have a long axis for the efficient absorption of incident sunlight, and short path lengths for the diffusion of charge carriers [103,106]. These nanostructures have been employed in applications beyond photocatalysis, ranging from solar cells [93,99,107] to sensors [108–113], but also in lithium–ion batteries [114–117], fuel cells [71] and so on.

## 2. Metal Oxide Photocatalytic Papers

$TiO_2$ is the most investigated metal oxide photocatalyst [81]; however, the use of other metal oxides in photocatalysis, such as ZnO, has been growing lately [30,31,118]. At the nanoscale, these materials can assume several structures, with direct influence on their photocatalytic activity. Moreover, the integration of such nanostructures in substrates is also challenging, including for cellulose-based substrates. The search for photocatalysts capable of performing the degradation of pollutants without the requirement of powder recovery gives visibility to the photocatalytic paper concept. In fact, the paper intrinsic structure can contribute to the photocatalytic degradation process with the possibility of being re-used over the time.

### 2.1. TiO₂ Photocatalytic Paper

Titanium dioxide has been largely investigated as a photocatalyst due to its strong oxidizing abilities for the decomposition of organic pollutants, and also its chemical and physical stabilities, nontoxicity, low cost, superhydrophilicity, and earth abundance [13,69,71,81]. $TiO_2$ occurs in nature, presenting three crystalline phases: two tetragonal phases, i.e., anatase and rutile, and an orthorhombic phase, i.e., brookite (Figure 3) [119]. However, it also occurs in an amorphous form. Anatase and brookite are both metastable phases that, when heated, transform into rutile, the most stable phase [120,121]. The lattice parameters of anatase ($I4_1/amd$) are $a$ = 0.3785 nm and $c$ = 0.9515 nm, for rutile ($P4_2/mnm$) are $a$ = 0.4594 nm and $c$ = 0.2958 nm, while brookite (*Pbca*) has lattice parameters of $a$ = 0.9184 nm, $b$ = 0.5447 nm, and $c$ = 0.5145 nm [122,123]. The three crystalline phases have $TiO_6$ octahedra with different distortions, in which the structural and electronic properties of all the phases can be determined by the Ti–O bonds. The description of the $TiO_2$ structure and stoichiometry is well documented elsewhere [13,71,122,124], so this will not be covered here.

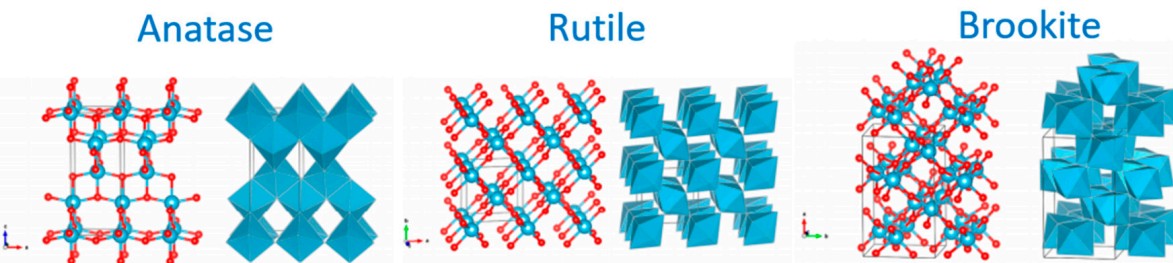

**Figure 3.** Unit cells of $TiO_2$ rutile, anatase and brookite [125]. Reprinted with permission from Structural Characteristics and Mechanical and Thermodynamic Properties of Nanocrystalline $TiO_2$, Chemical Reviews, 2014, 114, 19, 9613–9644. Copyright 2021 American Chemical Society.

$TiO_2$ is a *n*-type semiconductor material with a wide energy band gap, displaying optical band gaps of 3.0 and 3.2 eV for rutile and anatase, respectively [126]. The band gap values of brookite vary in the literature and have been reported to be from 3.13 to 3.40 eV [120,126]. With the large band gap of all $TiO_2$ phases, the solar radiation absorption is limited to the UV range, which restricts its widely photocatalytic application for environmental remediation. Other restrictions to the extensive use of $TiO_2$ in photocatalysis are related to the rapid recombination of the photogenerated electron–hole pairs and poor affinity towards hydrophobic organic pollutants [69].

$TiO_2$ is usually used as a photocatalyst in the forms of anatase, generally with higher photocatalytic activity, and rutile, which is less photoactive [48,69,127–130]. The enhanced photocatalytic performance of anatase can be related to three main reasons: (i) its larger band gap when compared to rutile, which raises the valence band maximum to higher energy levels relative to redox potentials of adsorbed molecules, thus influencing the oxidation of electrons and facilitating electron transfer from $TiO_2$ to adsorbed molecules; (ii) its indirect bandgap that exhibits a longer lifetime of photoexcited electrons and holes when compared to the direct bandgap of rutile and brookite [129]; and (iii) the concentration of oxygen vacancies is higher for anatase nanomaterials, which leads to a greater charge separation efficiency [121]. In the case of brookite, the interest on its photocatalytic activity has been growing lately [120,131,132], which has been reported to have higher photocatalytic activity than anatase or rutile [133]. The determination of the best photocatalyst among the three $TiO_2$ phases is still under intense debate. Moreover, it has been shown that the mixture of the $TiO_2$ phases displays higher photocatalytic activity than the pure phases [37,134], reducing the recombination of photogenerated holes and electrons [135]. In fact, the most common commercial photocatalyst is Degussa P-25, that consists of both rutile and anatase phases [130].

The higher photocatalytic activity in terms of exposed active facets was reported as {111} > {001} > {100} > {101} for anatase [136], {110} > {001} > {100} for rutile [129] and {210} for brookite [120]. The active facets have a direct effect on the surface adsorption or desorption abilities and the redox potential of the charge carriers, such as the surface area, with more active sites for reaction on the nanomaterials' surface [137].

To overcome the limitations involving the use of $TiO_2$ in photocatalysis under solar radiation, several approaches have been reported. One possibility involved the engineering of the $TiO_2$ optical bang gap by adding oxygen vacancies or doping with non-metal and metal elements to narrow its band gap [138–140]. It has been reported that oxygen vacancies facilitate the charge separation process [141], while doping with external elements can increase the redox potential of the radicals and increase quantum efficiency by reducing the degree of recombination of the electrons and holes, despite enlarging the $TiO_2$ absorption spectrum [142]. Surface modification, fabrication of composites or heterojunctions with other metal oxides or other materials such as graphene have also been reported to expand $TiO_2$ photocatalytic activity under solar radiation [69,143–147]. Lately, several reviews describing the $TiO_2$ band gap reduction to the visible region and further photocatalytic activity under visible/solar radiation have been conducted, demonstrating the use of different materials and several methods that were investigated [143,147–150].

$TiO_2$ is a very versatile material and to satisfy the most distinct applications of this material, different structures have been designed and produced. In terms of nanomaterials, $TiO_2$ nanowires, nanorods, nanofibers, nanotubes, nanobelts, nanowhiskers, nanospheres, and nanoflowers, among others, have been described (Figure 4) [36–38,151–156]. $TiO_2$ thin films have been also largely investigated [157–159]. Various techniques have been reported for producing $TiO_2$ nanostructures or thin films including wet-chemical techniques [160,161], the sol–gel method [84,158], thermal evaporation [162], magnetron sputtering [163,164], anodization [165,166], electrodeposition [167,168], the gas jet fiber spinning process [103], electrospinning [104,169], hydrothermal and solvothermal synthesis [170–173], and microwave irradiation [15,36,38], among several other methods.

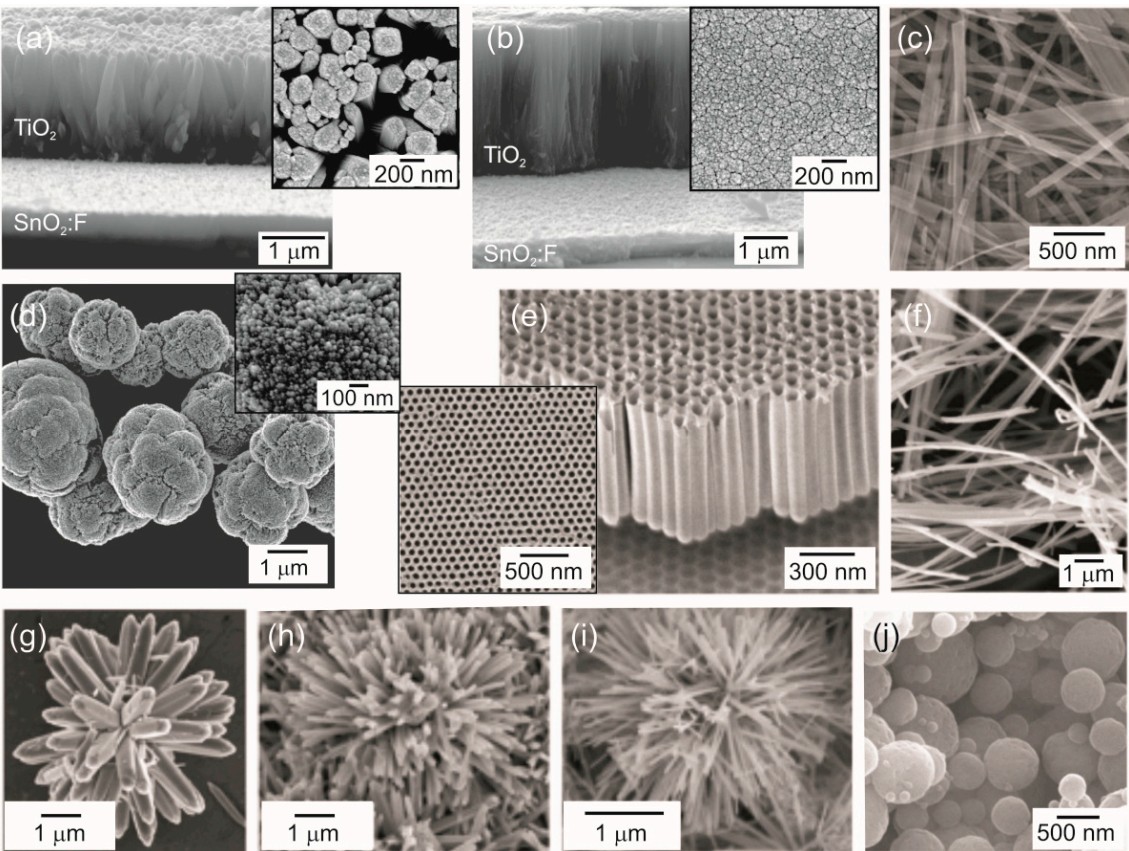

**Figure 4.** SEM images of several TiO$_2$ nanostructures. (**a**,**b**) TiO$_2$ nanorods grown of a SnO$_2$:F seed layers [38], (**c**) TiO$_2$ nanobelts [174], (**d**) TiO$_2$ nanorods aggregated to form micro-sized spheres [37], (**e**) TiO$_2$ nanotubes formed by anodization [175], (**f**) TiO$_2$ nanowires [176], (**g**–**i**) different types of TiO$_2$ nanoflowers [155], and (**j**) TiO$_2$ nanospheres [156]. Reproduced with permission of Elsevier [38,156,174–176], MDPI [37] and Springer Nature [155].

The integration of TiO$_2$ in cellulose-based materials to form composites or membranes has been reported, but the growth or deposition of thin films or nanostructures on a paper substrate has also shown promise [12,15,19,177,178]. Hu et al. [179] produced a flexible and binder-free graphene–TiO$_2$ paper to be applied as lithium–ion battery anode materials, consisting of 3-D nano-structures with nano-sized TiO$_2$ intercalated between graphene layers as pillars to increase the Li–ion insertion/extraction rate. In [15], electrochromic TiO$_2$ nanostructured films were grown on gold coated papers using a microwave-assisted hydrothermal method at a low synthesis temperature to produce electrochromic (EC) devices. The effect of the acid used in the microwave synthesis played a crucial role on the film's structure and final EC performance (Figure 5).

Several studies described the photocatalytic activity of TiO$_2$ paper-based photocatalysts with gaseous or liquid pollutants. In 1995, Matsubara et al. [180] reported the production of a TiO$_2$-containing paper and investigated its photocatalytic activity by measuring the decomposition of gaseous acetaldehyde under a weak UV light irradiation. Iguchi et al. [181] prepared photocatalytic papers containing TiO$_2$ supported on inorganic fibers. The photodegradation performance of acetaldehyde gas and the durability of the TiO$_2$-containing papers have been investigated under UV irradiation. In [182], photocatalytic papers fabricated with nano TiO$_2$ powders doped with nitrogen and iron and supported on X zeolite were tested for the photodegradation of acetaldehyde in air, as an indoor pollutant under visible light irradiation (batch conditions).

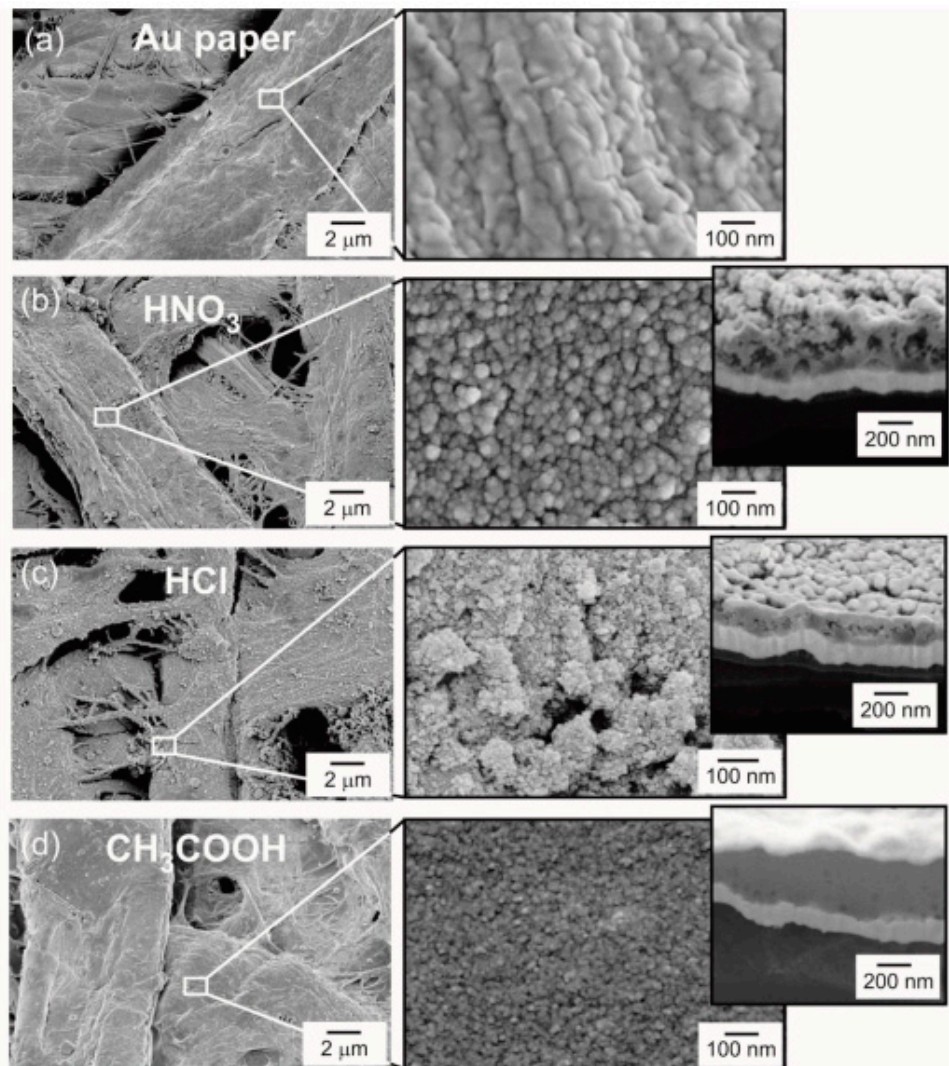

**Figure 5.** SEM images of the Whatman paper with an Au film and the TiO₂ nanostructured films grown under microwave irradiation to be used as EC devices. Magnified SEM image of the Au film in (**a**) and magnified and cross-section SEM images of the TiO₂ nanostructured films produced with (**b**) nitric acid; (**c**) hydrochloric acid; and (**d**) acetic acid [15]. Reproduced with permission of MDPI [15].

Sboui et al. [183] reported the functionalization of papers with TiO₂ decorated with AgBr nanoparticles. Their photocatalytic performance was evaluated through the degradation of 2-propanol in gas phase under sunlight exposure. In another approach, the fabrication of photocatalytic paper using TiO₂ nanoparticles confined in hollow silica capsules has been shown (Figure 6) [55]. The encapsulation demonstrated a shielding effect that can insulate the TiO₂ nanoparticles from the surrounding environment and prevent the self-degradation of organic support materials under UV light. The encapsulated TiO₂ nanoparticles have been deposited onto the cellulose paper either by a chemical adhesion process via ionic bonding or by a physical adhesion process using a dual polymer system. The photocatalytic activity was assessed with 2-propanol degradation under UV-light exposure.

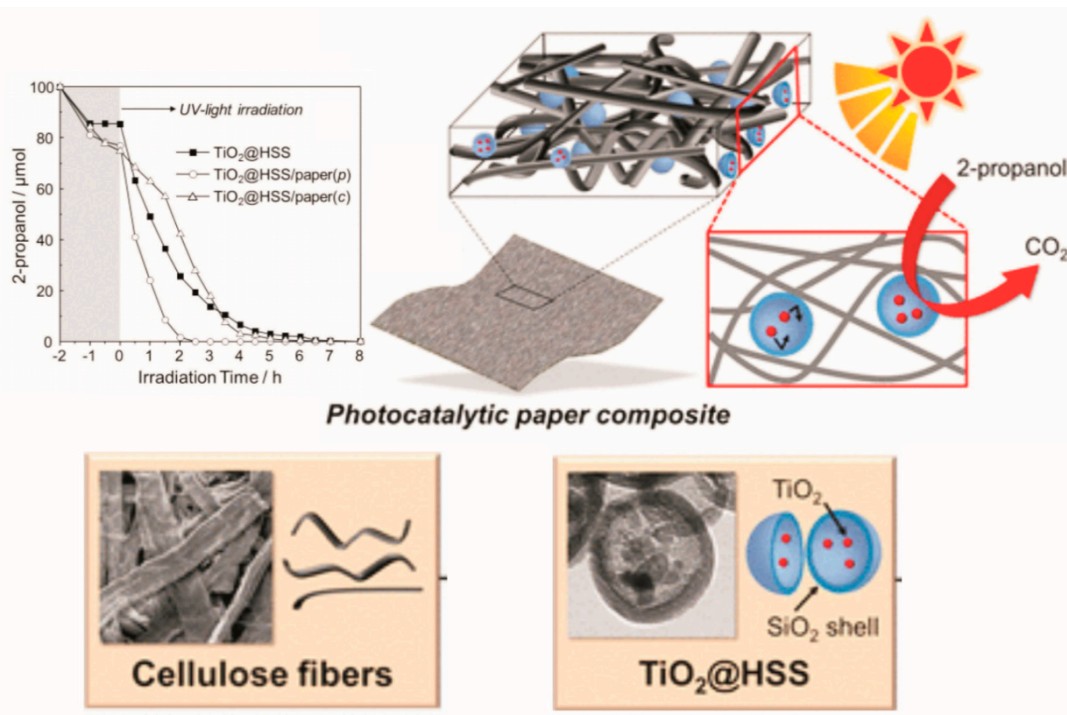

**Figure 6.** Photocatalytic degradation of 2-propanol and SEM images of paper fibers and the $TiO_2$ nanoparticles confined in hollow silica capsules (HSS) [55]. Reprinted with permission from fabrication of photocatalytic paper using $TiO_2$ nanoparticles confined in hollow silica capsules, Langmuir 2017, 33, 1, 288–295. Copyright 2021 American Chemical Society.

Zhang et al. [57] reported a cellulose-based photocatalytic material with $TiO_2$ nanoparticles loaded on carbon fibers. $Na_2SiO_3$ or $Al_2(SO_4)_3$ were used as adhesives, with differences in the photodegradation behaviors of methyl orange (MO) depending on the composite used. The ratio of carbon/cellulose fibers also demonstrated to influence the MO photodegradation. Qin et al. [58] described the production of photocatalytic papers containing a mixture of poly-dopamine-loaded cellulose fibers and pristine cellulose fibers and $TiO_2$ nanoparticles. The photocatalytic performance of $TiO_2$ photocatalytic paper was shown by decomposing methyl orange under UV light irradiation. In [184], $TiO_2$-coated non-woven paper was tested as a photocatalyst for the degradation of Rhodamine B in aqueous solutions. $TiO_2$ was coated on non-woven paper using an aqueous dispersion of colloidal $SiO_2$ binder. The photodegradation was temperature-dependent with a high degradation rate being achieved at high temperature. Moreover, the presence of $Cl^-$, $CH_3COO^-$ and $HPO_4^{2-}$ ions led to the reduction of the photodegradation rate, while the presence of $SO_4^{2-}$ increased it. The binder assists the adhesion of $TiO_2$ on paper fibers and prevents the degradation of paper imposed by the oxidation effect of $TiO_2$ [185]. Jiao et al. [186] showed anatase $TiO_2$ on paper and this photocatalyst degraded blue indigo carmine dye under UV radiation. Toro et al. [187] reported the fabrication of $TiO_2$ hydrosols on paper sheets and their photocatalytic behavior was investigated under UV radiation using methylene blue as a model dye.

Photocatalytic papers based on a $TiO_2$/Sodium alginate nanocomposite were reported in [185] (Figure 7a–c). Their photocatalytic activity was investigated with the removal of chemical oxygen demand of wastewater. The increase in the photocatalyst increased the wastewater mineralization and enhanced the removal of chemical oxygen demand, as expected (Figure 7d). It has been shown that the presence of sodium alginate as a biopolymer increased the adhesion of nanoparticles to paper fibers and reduced the harmful effect imposed by the photocatalyst.

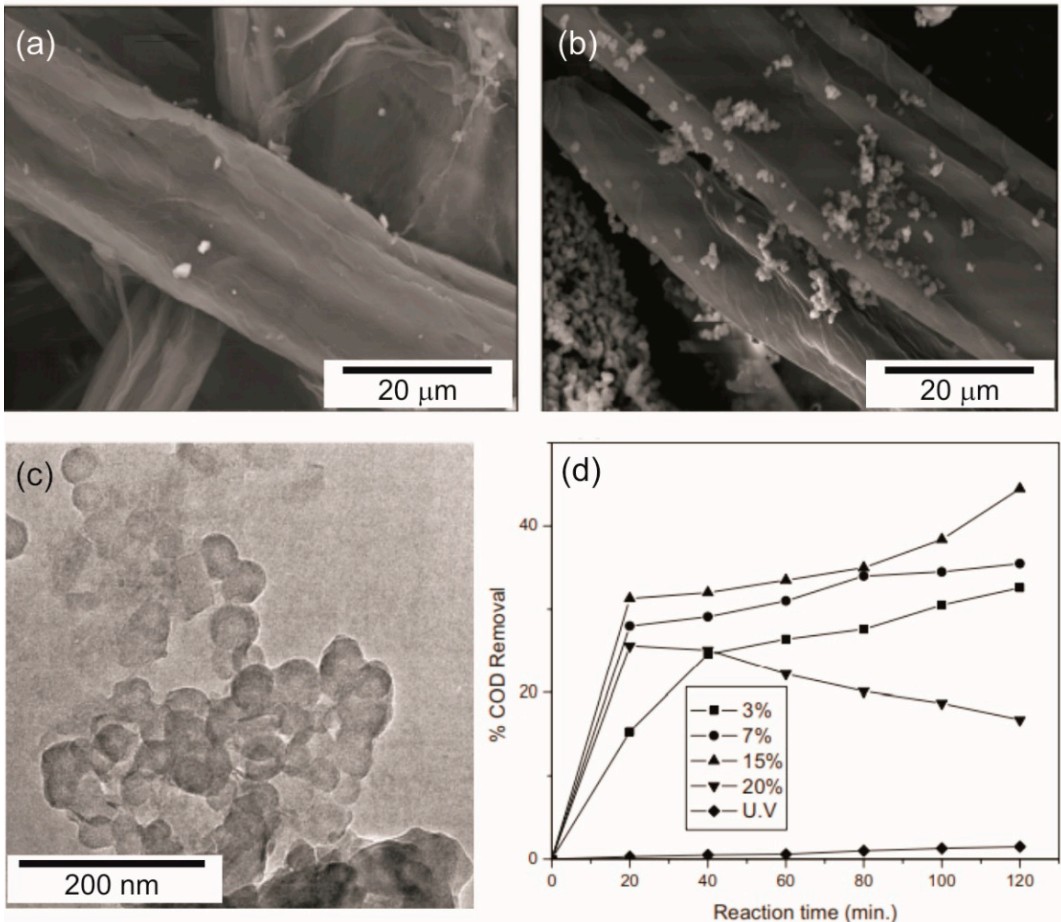

**Figure 7.** (**a**) SEM images of the photocatalytic paper containing 7% and (**b**) 20% of TiO$_2$/Sodium alginate nanocomposite, and (**c**) transmission electron microscopy (TEM) images of the nanocomposite. (**d**) Photocatalytic degradation for the removal of chemical oxygen demand of wastewater. Reproduced with permission of Elsevier [185].

A paper-TiO$_2$ composite was developed in [178] and the efficiency of the resulting photocatalyst was investigated by degradation of gaseous 2-propanol under simulated sunlight irradiation. Another study reported a ternary composite composed by Au nanoparticles decorating TiO$_2$ nanowires onto hierarchically porous carbon fiber paper. Its photocatalytic activity was investigated with the degradation of gaseous styrene under visible light irradiation [177].

Two analogous studies reported the production of photocatalytic papers composed by nanosized TiO$_2$ supported on natural zeolite [188,189]. The fibers formed a closely packed network with microvoids, with the zeolite-based TiO$_2$ particles being randomly attached to these fibers. The natural zeolite-based TiO$_2$ composite sheets decomposed gaseous toluene under UV radiation, and the photocatalytic efficiency was attributed to the synergy effect with (1) confinement of the organic contaminant in the microvoids between the fiber networks, (2) its further adsorption on zeolite and (3) the subsequent photodecomposition of adsorbate by TiO$_2$ nanoparticles [188]. In [190], a photocatalytic paper formed by a composite of TiO$_2$/SiO$_2$ particles introduced in bulk paper had its activity demonstrated by the degradation of ethanol (50–200 ppm) under UV irradiation.

A review on the topic was published in 2006, where Pelton et al. [56] revised the photocatalytic paper developments at that time, which included methods to fix TiO$_2$ on cellulose substrates to minimize photochemical damage to the paper; moreover, the use of multiple approaches for enhancing mineralization were discussed for better photocatalytic disinfection.

When it comes to growing TiO$_2$ nanostructured films on paper, where paper is the substrate, D. Nunes et al. [19] reported the synthesis of TiO$_2$ nanostructured films using microwave irradiation at low temperature synthesis (80 °C) and without any seed layer, in which bacterial nanocellulose and tracing paper were tested as substrates (Figure 8a–d). Their photocatalytic activity was investigated using Rhodamine B degradation under solar radiation. The nanocellulose based material demonstrated higher photocatalytic activity than tracing paper, in which the structural differences of the TiO$_2$ nanostructured films and substrates play a key role on the behavior observed (Figure 8e,f). It has been reported that a nanocellulose substrate with a 3D structure at the nanoscale can effectively enhance photocatalytic activity. Moreover, the 3D closed packed cellulose nanofiber network of nanocellulose fully covered with TiO$_2$ provides more active sites for the photoreaction, and also facilitates the species' transport and electrons' collection [191]. The BNC-based material could be reutilized despite activity deterioration over the exposures.

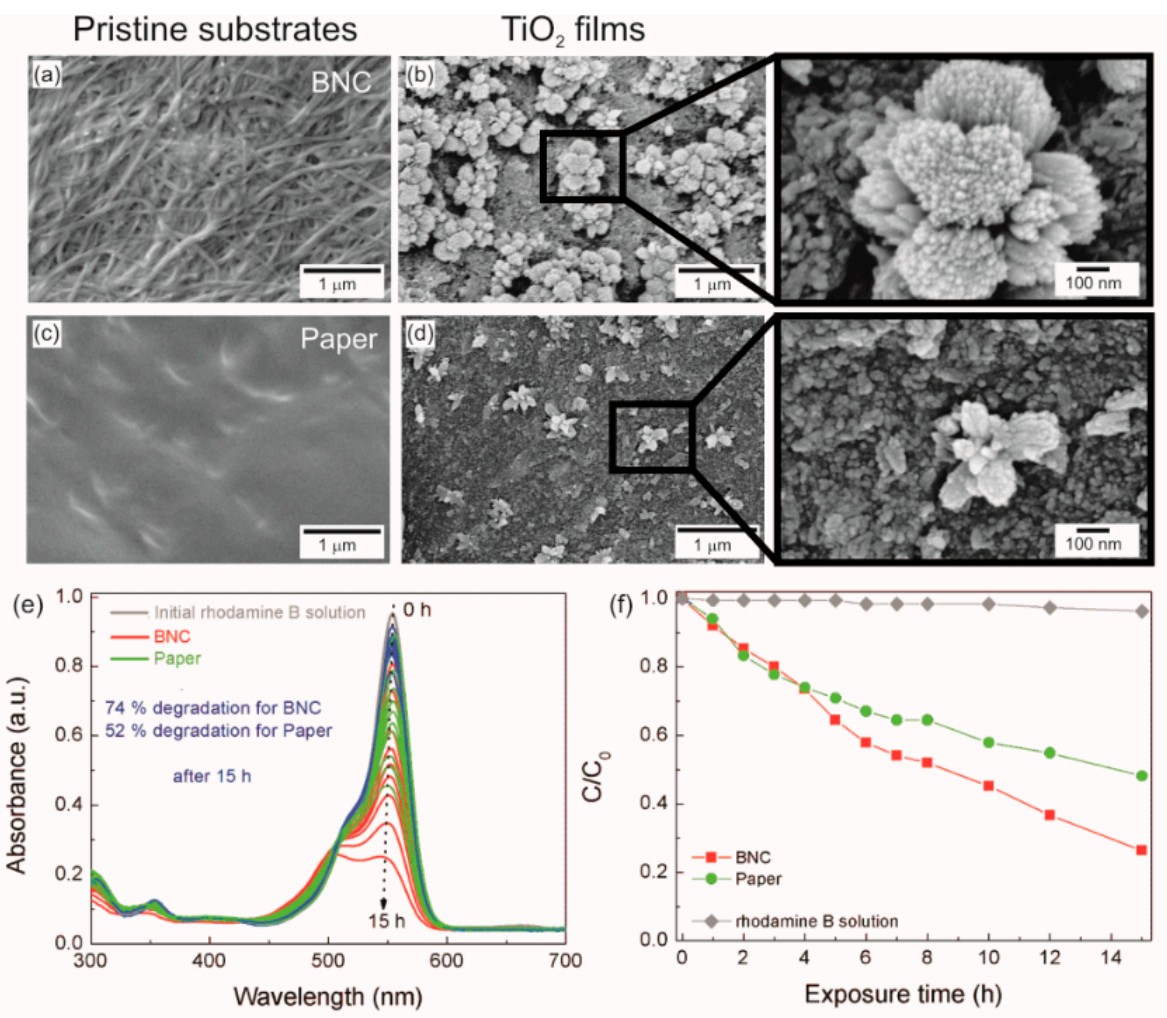

**Figure 8.** SEM images showing the BNC (**a**) and tracing paper (**c**) pristine substrates together with the TiO$_2$ films grown on BNC (**b**) and tracing paper (**d**) substrates. The insets show the TiO$_2$ films and the nanorod flower-like structures. Rhodamine B absorbance spectra at different solar light exposure times up to 15 h (**e**). Rhodamine B degradation ratio (C/C$_0$) vs. solar simulating light exposure time (**f**). Reproduced with permission of *Springer Nature* [19].

Office paper was also used as substrates for growing TiO$_2$ nanostructured films. Figure 9 showed the scanning electron microscopy (SEM) images of the pristine office paper (Figure 9a) and the TiO$_2$ nanostructured films grown under microwave irradiation on the office paper substrate (Figure 9b). It can be observed that microwave synthesis totally covered the paper substrate, and a continuous nanostructured TiO$_2$ film could be

achieved. These films were investigated as photocatalysts, and their photocatalytic activity was assessed for Rhodamine B degradation under UV radiation (Figure 9c). The gradual rhodamine B degradation was observed. The photocatalysts were resistant to water immersion, and after the photocatalytic experiments, the papers could be recovered and dried for further tests. The direct growth of $TiO_2$ on paper with chemical-based processes reduces the production costs of the photocatalysts due to the lack of parallel processes, such as seed layers, or annealing. An analogous study reported the photocatalytic degradation of Rhodamine B using office paper as a substrate under UV and solar radiation [12]. This study demonstrated the synthesis of $TiO_2$ and ZnO nanostructures using hydrothermal synthesis assisted by microwave irradiation, using office, Whatman, and commercial hospital papers as substrates. Their photocatalytic activity was assessed, and higher Rhodamine B degradation was reported for photocatalysts grown on Whatman paper [12]. T. Freire et al. [192] reported the growth of $TiO_2$ thin films composed by very fine particles of ~11 nm on Whatman paper using hydrothermal synthesis assisted by microwave irradiation. These films had their photocatalytic activity tested with the photodegradation of Rhodamine B under solar radiation, and the effective contribution of paper on the final photocatalytic performance has been estimated.

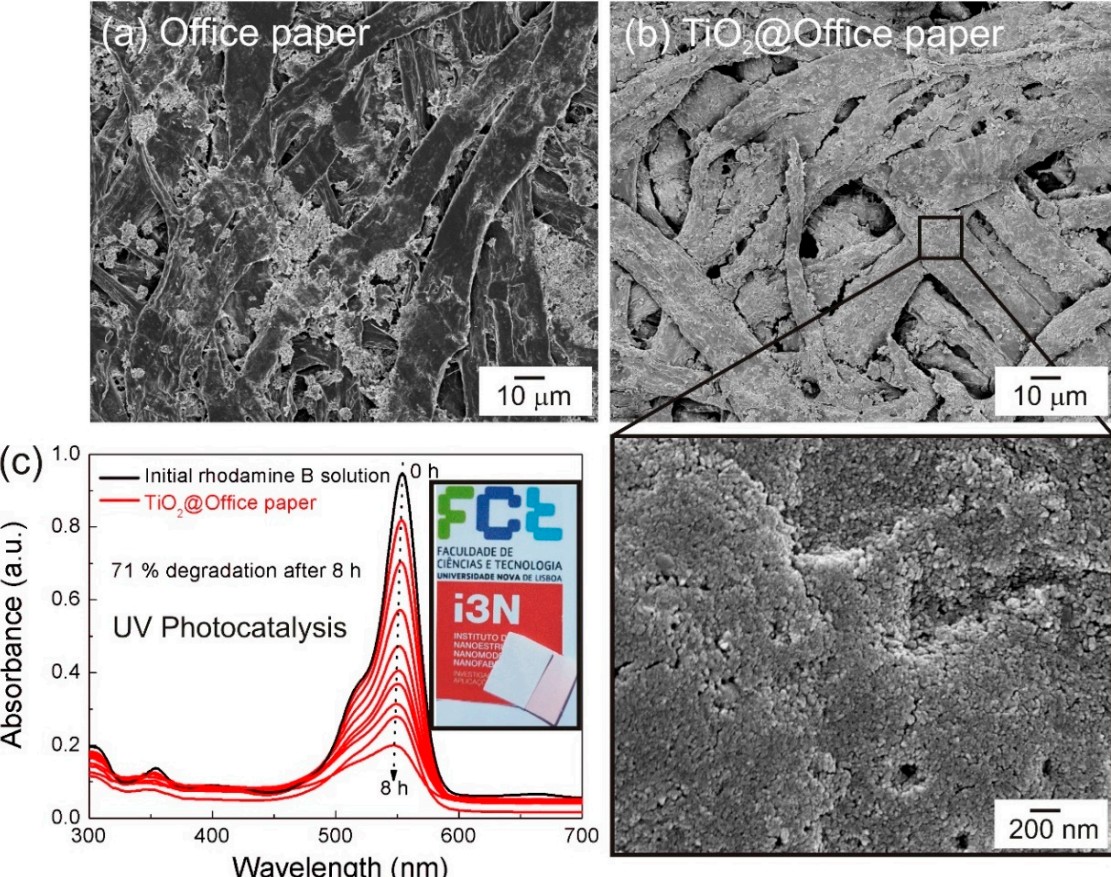

**Figure 9.** SEM images showing (**a**) the pristine office paper and (**b**) after a microwave-assisted hydrothermal synthesis. The inset magnifies the $TiO_2$ nanostructured film grown on office paper substrate. (**c**) Rhodamine B absorbance spectra at different UV exposure times using the $TiO_2$ nanostructured film on office paper together with the image of the photocatalytic paper before and after (pinkish color) experiments (71% of degradation after 8 h using 3 lamps of 95 W aligned in parallel, from Osram, with an emission wavelength of 254 nm, and Rhodamine B solution of 5 mg/L). Photocatalytic experiments based on [19].

*2.2. ZnO Photocatalytic Paper*

Zinc oxide is also considered a promising photocatalyst for environmental remediation. In fact, it has been reported that ZnO exhibits higher efficiency in the photocatalytic degradation of some organic dyes than $TiO_2$, i.e., methyl orange and Rhodamine 6G [193]. ZnO has a high surface reactivity owing to its large number of active surface defect states [20]. This material has a strong oxidation ability, chemical stability, enhanced photocatalytic activity, and a large free-exciton binding energy, in a way that the excitonic emission processes can be maintained at room temperature or above [30]. Moreover, it is non-toxic, earth abundant, biocompatible, biodegradable, environmentally friendly, low cost, and compatible with simple chemical synthesis. Just like $TiO_2$, its wide band gap limits light absorption in the visible light region which is associated with the fast recombination of photogenerated charges, resulting in low photocatalytic efficiency [30].

ZnO is an *n*-type semiconductor with a direct and wide band gap of 3.37 eV and a large exciton binding energy of 60 meV at room temperature [30,194]. The common ZnO crystal structures are the rocksalt, wurtzite or cubic (zinc blende) structures. However, ZnO displays the wurtzite crystal structure at room temperature (Figure 10), with the highest thermodynamic stability among the three structures [30,195]. This wurtzite structure has a hexagonal unit cell with space group *P6₃mc* and lattice parameters, *a* = 0.3296 nm and *b* = 0.52065 nm [30]. The ZnO wurtzite structure has a tetrahedrally coordinated bonding geometry based on two interconnecting hexagonal closed packed sublattices, each of them consisting of one type of atom (Zn or O) displaced alternatively along the threefold *c*-axis. It exhibits a positive polar plane that is rich in $Zn^{2+}$ and negative polar plane that is rich in $O^{2-}$. Each zinc ion has four oxygen neighbor ions in a tetrahedral configuration and vice versa [30,196]. The tetrahedral coordination is typical of $sp^3$ covalent bonding, and this $Zn^{2+}$ and $O^{2-}$ tetrahedral coordination is the origin of a polar symmetry along the hexagonal axis. Moreover, the polarization effect is one of the major factors influencing the crystal growth of ZnO nanostructures [30].

The *c*-axis direction is referred to as [0001], while the surface perpendicular to the *c*-axis is the hexagonal (0001) plane [196]. The most common and stable ZnO wurtzite crystal exhibits the following face terminations: the polar Zn terminated {0001} and O terminated {000$\bar{1}$} facets, and the non-polar {10$\bar{1}$0} facets, containing an equal number of Zn and O atoms [62,194,197–199]. The polar facets possess distinct chemical and physical properties when compared to the non-polar ones [194].

It has been reported that the {0001} facets terminated with Zn atoms are the most active facets among the various surfaces of ZnO nanomaterials. Thus, due to the high surface energy of {0001} facets, the exposure of {0001} facets may enhance the efficiency of photocatalysis. However, it has been shown that nanostructures with high percentages of exposed high-index facets also exhibited superior photocatalytic activity [200].

ZnO has also restrictions to its widespread use in photocatalysis under solar radiation as previously mentioned. Thus, several approaches have been suggested to overcome this limitation, including nonmetal and metal doping for reducing the band gap and improving the charge carrier separation, which shifts the absorption range of ZnO to the visible region [118]. Moreover, the surface functionalization of ZnO also has an impact on the final photocatalytic performance due to the narrowing of the material's surface band gap [201]. Another approach lies on making an enhanced heterojunction with a *p*-type semiconductor, for example, CuO, to achieve superior photocatalytic activity [202].

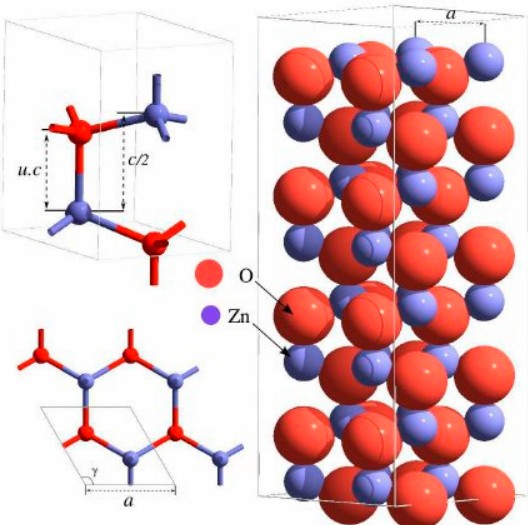

**Figure 10.** Bulk crystal structure of wurtzite Zinc oxide [203]. Reprinted Figure with permission from Simulation of reconstructions of the polar ZnO {0001} surfaces, as follows: H. Meskine and P. A. Mulheran, Phys. Rev. B 84, 165430, 2011. Copyright 2021 by the American Physical Society.

ZnO has been extensively investigated over the years as thin films or nanostructures and in fact, several ZnO structures have been described at the nanoscale. The most common are one-dimensional nanostructures, such as nanorods, nanotubes, nanofibers, nanowires, but also nanoplates, nanosheets, nanospheres, tetrapods, and nanoflowers, among others, have been described (Figure 11) [204–209]. To produce such a variety of structures, numerous and distinct techniques have been reported, including direct precipitation [210], the sol–gel method [211,212], hydrothermal [213,214] and solvothermal syntheses [101], microwave synthesis [96,97,215], chemical bath deposition [216], electrospinning [217], electrodeposition [218], electrospinning [102,105], magnetron sputtering [219,220], and spray pyrolysis [221], among others.

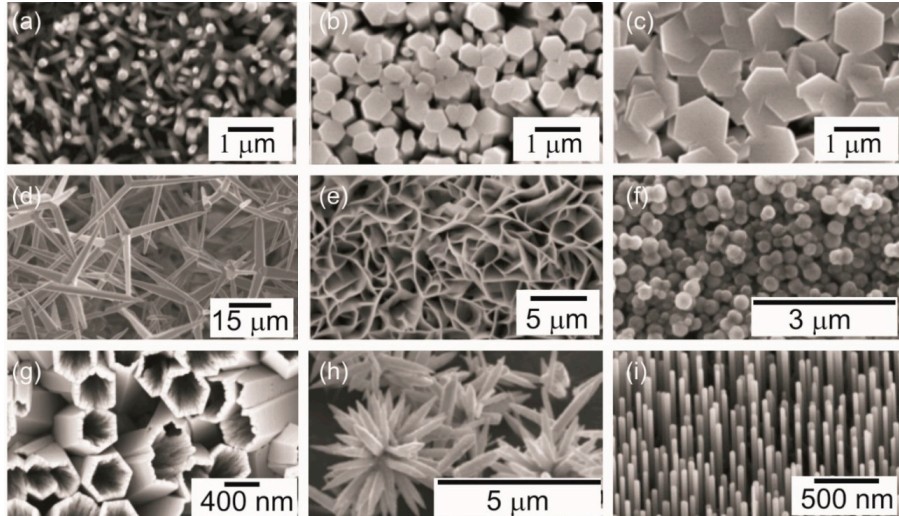

**Figure 11.** SEM images of several ZnO nanostructures. (**a**) ZnO nanorods [204], (**b**,**c**) ZnO nanoplates [204], (**d**) ZnO tetrapods [222], (**e**) ZnO nanosheets [205], (**f**) ZnO nanospheres [206], (**g**) ZnO nanotubes [207], (**h**) ZnO nanoflowers [209], and (**i**) aligned ZnO nanowire arrays [208]. Reprinted with permission from direct growth of freestanding ZnO tetrapod networks for multifunctional applications in photocatalysis, UV photodetection, and gas sensing, ACS Appl. Mater. Interfaces 2015, 7, 26, 14303–14316, copyright 2021 American Chemical Society [222], and Elsevier [204–209].

The growth of ZnO on paper substrates has also been reported previously. Araujo et al. [223] produced 3D SERS platforms based on ZnO nanorod arrays grown on paper substrates using a fast and low-temperature hydrothermal method assisted by microwave radiation. Pimentel et al. [11] produced paper-based UV sensors based on ZnO nanorods, also grown using a fast and low-temperature hydrothermal method assisted by microwave radiation. Two types of paper substrates were tested, i.e., tracing and Whatman papers. The effect of the synthesis temperature on ZnO nanostructures was investigated and an UV/Ozone treatment performed directly to the ZnO seed layer prior to microwave assisted synthesis revealed expressive differences regarding the formation of the ZnO nanostructures (Figure 12). Manekkathodi et al. [224] reported the production of prototype photoconducting devices and PN junction diodes were fabricated with aligned single-crystal ZnO nanowires and nanoneedles on paper using low temperature and a non-hazardous chemical solution.

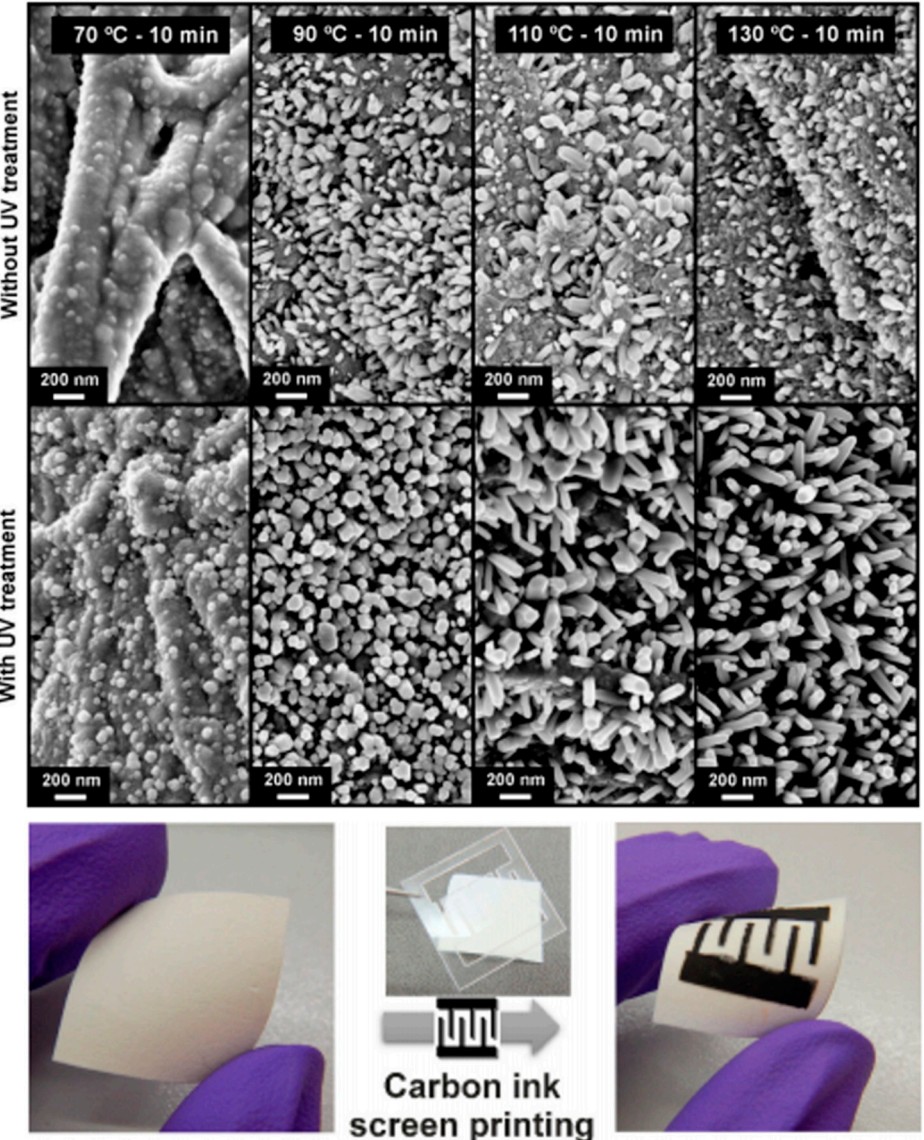

**Figure 12.** SEM images of the ZnO nanorods grown on Whatman paper with different temperatures in microwave synthesis. The effect of UV treatment has been investigated. The real image of the paper-based UV sensor is presented [11]. Reproduced with permission of MDPI [11].

The antibacterial activity against *Escherichia coli* of paper coated with ZnO nanoparticles has been reported in [225]. In another study, ZnO-cellulose composites coating paper

were applied as antibacterial and antifungal agents [226]. A review manuscript dedicated to the preparation and antibacterial activity of cellulose/ZnO composites has been published in [227].

When it comes to ZnO photocatalytic paper, some studies demonstrated the enhanced photocatalytic activity of such metal oxide material on paper, especially of 1D nanostructures, such as nanorods, nanowires, and nanofibers, due to their larger surface-to-volume ratio, when compared to thin films [20].

Baruah et al. [20] demonstrated that ZnO nanorods grown on paper supports prepared from soft wood pulp are promising photocatalysts. The produced paper embedded with ZnO nanorods in its porous matrix revealed enhanced photocatalytic degradation of methylene blue and methyl orange under visible light, being reused several times. Tsai et al. [54] reported a photocatalytic paper comprised of $Cu_2O$ and Ag nanoparticles decorating ZnO nanorods produced using a hydrothermal method, and its application in the visible light photodegradation of Rhodamine B. The photocatalytic activity of pure ZnO-based paper and paper containing $Cu_2O$ and Ag separately or together have been tested, in which the paper with $Cu_2O$ and Ag nanoparticles co-decorating the ZnO nanorods demonstrated the best photocatalytic efficiency (Figure 13).

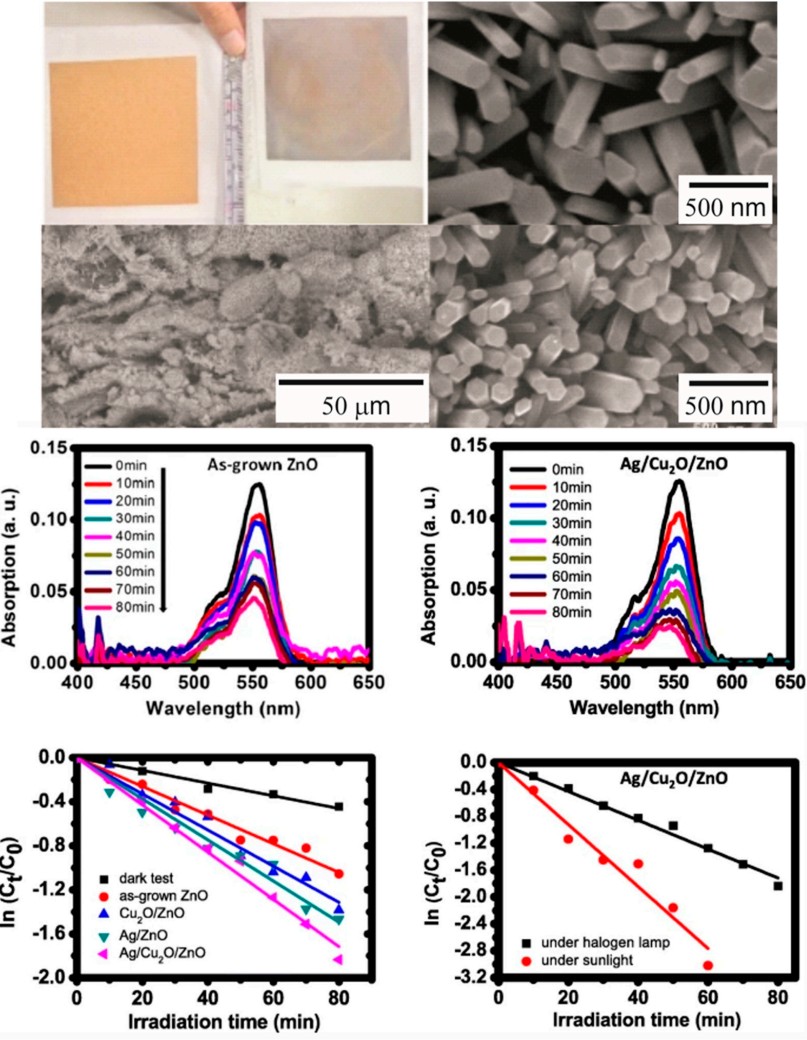

**Figure 13.** Real photographs of the kraft paper substrate and the paper with the ZnO nanorods. The SEM image of the papers containing the ZnO nanorods are presented together with their photocatalytic activity under visible light [54]. Reproduced with permission of Springer Nature [54].

A thorn-like ZnO-multiwalled carbon nanotube hybrid paper produced with an atomic layer deposition followed by hydrothermal growth has been reported in [228]. Its photocatalytic performance was assessed under UV irradiation, enabling the photodegradation of Rhodamine B and good recyclability.

In an innovative approach, $Fe_2O_3/ZnO$ hollow spheres with submicron pores were integrated in inks to produce stable photocatalytic papers by a spraying method. It was reported that the hollow spheres perfectly matched the cellulose networks within the paper due to the similar dimensions, and that the photocatalytic paper were able to efficiently degrade the 2,4,6-trichlorophenol under solar light irradiation [229].

A heterojunction of $TiO_2/ZnO$ nanostructures grown on paper substrates has been reported in [12]. Microwave synthesis was used for growing both layers and depending on the type of paper used, different ZnO structures were obtained (Figure 14). Continuous ZnO nanorod arrays were grown on Whatman paper, while on office paper, nanoplates that originated the nanoflower structures could be observed. The formation of nanoplates structures have been associated with the presence of calcium carbonate ($CaCO_3$) in office paper. The effect of oxalic acid has also been investigated, and an etching effect was observed on office paper with deterioration of the nanoplate's surface and laminar structures, and holes started to appear. The $ZnO/TiO_2$ heterostructures grown on office paper were tested as a photocatalyst under UV light. It was shown that the deteriorated structure of the nanoplates increased the photocatalytic activity due to the higher surface area of such materials.

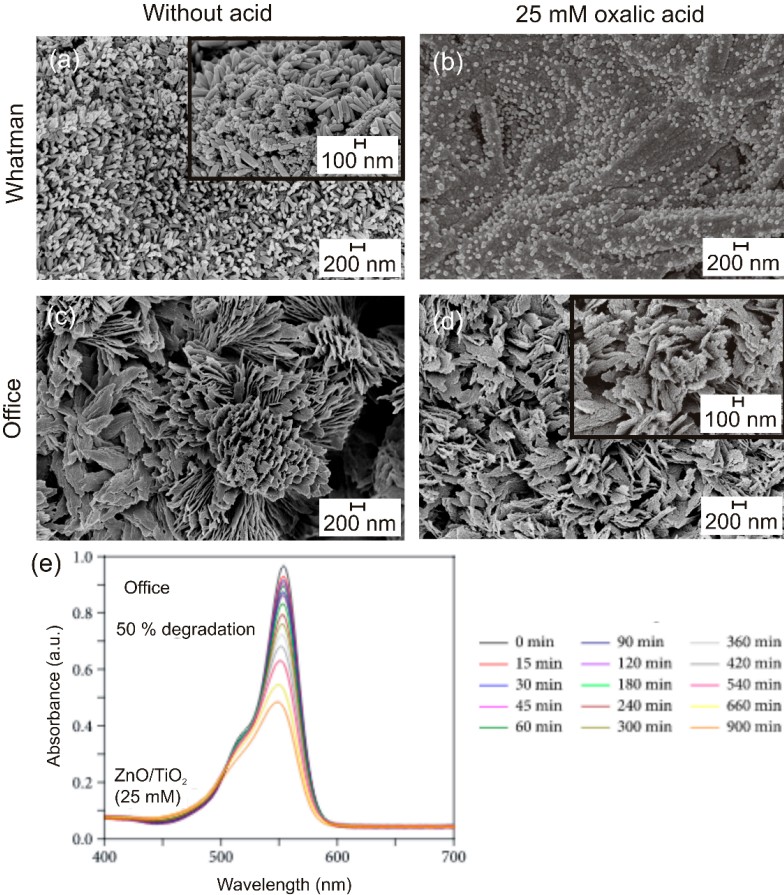

**Figure 14.** $ZnO/TiO_2$ heterostructures grown on Whatman and Office papers having $TiO_2$ synthesized without acid and with 25 mM of oxalic acid (**a**–**d**). The inset in (**a**) magnifies the heterostructure, while the inset in (**d**) evidences the surface modification with oxalic acid. (**e**) Rhodamine B photodegradation under UV light of the heterojunction grown on office paper and represented in (**d**) [12]. Reproduced with permission of Hindawi [12].

Other metal oxide-based papers had their photocatalytic activity investigated for environmental remediation; however, these studies are still scarce. A photocatalytic paper based on $Ag_2O$ has been reported, in which cellulose fibers and graphite fibers have been incorporated with the $Ag_2O$ nanoparticles. Its photocatalytic activity was evaluated with the degradation of methyl orange under UV, VIS or near-IR light, mostly covering the solar spectrum [230]. Another study reported a composite based on $BiVO_4$@diatomite/microcrystalline cellulose/poly (vinyl butyral) that was prepared using the electrospinning method. The photocatalytic activity of this paper was investigated with the photodegradation of methylene blue and formaldehyde under visible light [231].

The International Standard ISO 10678:2010 is an accepted standard experimental method for photocatalytic experiments and it specifies a method for the photocatalytic degradation of methylene blue [232]. Nevertheless, in most published studies, the photocatalytic parameters that can influence the final photodegradation vary expressively, including the light sources, the concentration of pollutants tested, and even the distance between the light source and the photocatalytic apparatus. Table 1 summarizes all the metal oxide-based photocatalytic papers discussed above, considering the paper-based materials, target pollutant, and light source.

**Table 1.** Summary of the metal oxide-based photocatalytic papers reported in literature.

| Material | Target Pollutant/Medium | Light Source | Reference |
|---|---|---|---|
| Paper with $TiO_2$ aggregates | Acetaldehyde in gas | Weak UV | [180] |
| Paper with $TiO_2$ supported on inorganic fibers | Acetaldehyde in gas | UV | [181] |
| Paper with nano $TiO_2$ powders supported on X zeolite | Acetaldehyde in gas | VIS | [182] |
| Papers containing $TiO_2$ decorated with AgBr nanoparticles | 2-propanol in gas | Sunlight | [183] |
| Paper with $TiO_2$ nanoparticles confined in hollow silica capsules | 2-propanol in gas | UV | [55] |
| Cellulose-based material with $TiO_2$ nanoparticles loaded on carbon fibers | Methyl orange in liquid | UV | [57] |
| Papers containing poly-dopamine-loaded cellulose fiere together with pristine cellulose fiere and $TiO_2$ nanoparticles | Methyl orange in liquid | UV | [58] |
| $TiO_2$-coated non-woven paper with colloidal $SiO_2$ binder | Rhodamine B in liquid | UV | [184] |
| Paper with $TiO_2$ nanosol | Blue indigo carmine in liquid | UV | [186] |
| $TiO_2$ hydrosols on paper sheets | Methylene blue in liquid | UV | [187] |
| Paper based on a $TiO_2$/Sodium alginate nanocomposite | Chemical oxygen demand of wastewater/liquid | UV | [185] |
| Paper-$TiO_2$ composite | 2-propanol in gas | Sunlight | [178] |
| Au nanoparticles decorating $TiO_2$ nanowires onto hierarchically porous carbon fiber paper | Styrene in gas | VIS | [177] |
| Papers composed by nanosized $TiO_2$ supported on natural zeolite | Toluene in gas | UV | [188,189] |
| Paper formed by a composite of $TiO_2$/$SiO_2$ particles | Ethanol in gas | UV | [190] |
| BNC with $TiO_2$ nanostructured films | Rhodamine B in liquid | Sunlight | [19] |
| Papers with $TiO_2$, ZnO and ZnO/ $TiO_2$ nanostructured films | Rhodamine B in liquid | UV and sunlight | [12] |

**Table 1.** *Cont*.

| Material | Target Pollutant/Medium | Light Source | Reference |
|---|---|---|---|
| Papers with $TiO_2$ thin films | Rhodamine B in liquid | Sunlight | [192] |
| Paper with ZnO nanorods | Methylene blue and methyl orange in liquid | VIS | [20] |
| Paper with $Cu_2O$ and Ag nanoparticles decorating ZnO nanorods | Rhodamine B in liquid | VIS | [54] |
| Thorn-like ZnO-multiwalled carbon nanotube hybrid paper | Rhodamine B in liquid | UV | [228] |
| Papers with $Fe_2O_3$/ZnO hollow spheres | 2,4,6-trichlorophenol in liquid | Sunlight | [229] |
| Paper containing $Ag_2O$ nanoparticles | Methyl orange in liquid | UV, VIS or near-IR | [230] |
| Paper based of $BiVO_4$@diatomite/microcrystalline cellulose/ poly(vinyl butyral) | Methylene blue in liquid and formaldehyde in gas | VIS | [231] |

All the approaches discussed in this review manuscript provide an overview on the versatility of using cellulose-based materials on photocatalysis. However, these materials still present some drawbacks, such as degradation under light exposure or the presence of impurities. An alternative to surpass these limitations is surface functionalization. In fact, functionalization strengthens thermal, mechanical and barrier properties [1] and increases surface absorption which, in the case of photocatalysis, influences the final performance. Moreover, it can create hydrophobic barriers or regions to contain compounds/reagents or pollutants. Cellulose functionalized with several binders (porphyrin, phthalocyanine, polyaniline (PAni)), and cellulose embedded with silver nanowires have been previously reported [1,233,234]. Thus, the integration of enhanced metal oxides as membranes, composites or grown/deposited in functionalized cellulose-based materials can originate highly photoactive papers to help environmental remediation.

## 3. Conclusions and Future Perspectives

This review summarized the latest developments in photocatalysis, focusing on metal oxides integrated on cellulose-based materials to originate enhanced photocatalytic papers. The work described the use of paper as a substrate, but also to form composites and membranes. The main characteristics of paper have been emphasized, including its low-cost and abundant character; moreover, it is environmentally friendly, flexible, foldable, recyclable, and lightweight. For photocatalysis, its 3D structure can contribute to enhance the photocatalytic activity, providing more active sites for the photoreaction.

Metal oxide nanomaterials are largely employed in photocatalysis, due to their high surface-to-volume ratios and high surface reaction activities. This work focused on nanostructured $TiO_2$ and ZnO since both materials are largely employed in the photodegradation of organic compounds. Moreover, these materials are eco-friendly, earth abundant, inexpensive, nontoxic, easily produced, and compatible with wet-chemical synthesis. Different studies have been presented which have correlated structural properties to the final photocatalytic activity. The main photocatalytic limitations of these materials have been addressed, namely the fast recombination of electrons and holes and the limitation of solar spectrum absorption, with different approaches for overcoming those being discussed. Other metal oxide-based paper systems and their applications for environmental remediation have also been mentioned.

The future of photocatalytic paper is expected to rely on nanosized cellulose. It is imperative to develop sustainable and inexpensive production processes for nanocellulose and its scale-up to industrial levels. To improve the photocatalytic activity of nanocellulose, surface functionalization is critical. The cellulose functionalization also allows stacking different paper-based layers: for example, a composite of a *n*-type semiconductor/cellulose

on top of a *p*-type semiconductor/cellulose (or vice versa), forming an enhanced photocatalytic heterojunction. Advanced 3D materials composed by a mixture of nano and micro-sized cellulose can also be relevant. The use of printing techniques is also an alternative to produce continuous photocatalytic films on nano-paper by directly printing inks containing the metal oxide nanostructures.

The increase in photocatalytic activity involves the development of better photocatalysts, i.e., metal oxide nanostructures or thin films. Several approaches have been discussed, from metal oxides decorated with nanoparticles, to doping with other elements, or even coupling with other semiconductors. However, extensive efforts are still needed for the development of innovative production strategies and tuned nanomaterials which are highly photoactive under solar radiation. Moreover, the deep understanding of the degradation mechanisms is still required.

The photocatalytic paper concept opens a wide number of possibilities, as these materials can be employed in varied applications, are easily adapted to different surfaces due to their high flexibility and have easy handling. Their disposable character and recyclability effectively contribute to environmental protection, while reducing costs.

**Author Contributions:** D.N. and A.P. were responsible for writing the review manuscript, R.B. participated in the review and editing, and R.M. and E.F. were responsible for supervising all the process. All authors have read and agreed to the published version of the manuscript.

**Funding:** This work was funded by National Funds through the FCT-Fundação para a Ciência e a Tecnologia, I.P., under the scope of the project UIDB/50025/2020–2023. The authors also acknowledge Fundação para a Ciência e a Tecnologia for funding the Project ICARUS under the reference PTDC/EAM-AMB/30989/2017. The work was also partially funded by the Nanomark collaborative project between INCM (Imprensa Nacional-Casa da Moeda) and CENIMAT/i3N. This work also received funding from the European Community's H2020 program under grant agreement No. 787410 (ERC-2018-AdG DIGISMART). The authors also acknowledge the funding of EC project SYNERGY H2020-WIDESPREAD-2020-5, CSA, proposal nº 952169.

**Conflicts of Interest:** The authors declare no conflict of interest.

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
