# Peer review of "Metal Oxide-Based Photocatalytic Paper: A Green Alternative for Environmental Remediation"

_catalysts, doi:10.3390/catal11040504_

Round 1
Reviewer 1 Report
Please see the attachment.

Author Response
Dear reviewer,
The authors acknowledge the pertinence of your comments and have revised the paper by taking them into account. The review manuscript has been completely revised and the English has been improved by R. Branquinho, and for that, she has been added as author. All the changes are highlighted. Following the reviewer’s comments order:
Referee 1:
Comment 1:
Summary: In this review, authors talk about recent developments of photocatalytic papers with nanostructured semiconducting metal oxide for environmental remediation. They particularly cover different characteristics of semiconducting metal oxides like titanium dioxide (TiO2) and zinc oxide (ZnO) in the form of nanostructures or thin films and their photocatalytic activity. The review also presents the role of cellulose-based materials in creating an enhanced photocatalytic paper with nanostructured TiO2 and ZnO. The review appears to be interesting, and the content of the review could be valuable to the readers of this journal. However, the current manuscript has some major drawbacks. Many minor issues are also present. Authors need to address those issues as illustrated in below comments. Hence, the manuscript is recommended for a major revision.
Comment 1: Authors need to take care of the following (see yellow highlight) – L12:to water splitting, CO2 and N2 reductions..”
- The authors acknowledge the reviewer’s comment, and the change has been carried out.
L17: … the use of metal….” Write Semiconducting metal oxides and authors should be consistent in terms of using metal oxide or semiconducting metal oxides.
- The authors acknowledge the reviewer’s comment, and all the manuscript has been changed and the term “metal oxide” has been selected. The word semiconductor was used for describing the photocatalytic process.
L22 titanium dioxide (TiO2)…This review manuscript explores…..” write “review”
- The authors acknowledge the reviewer’s comment, and the change has been carried out.
Check out the sentence “…most abundant ….gradable, earth abundant and….”
- The authors acknowledge the reviewer’s comment, and the term earth abundance has been removed.
L42: units (linear chain of ringed glucose units),….” –
- The authors acknowledge the reviewer’s comment, and the sentence has been changed. Please refer to page 41.
check out the format from the van der 47 Walls – should be “Van”
- The authors acknowledge the reviewer’s comment, and the change has been carried out.
“Goals by the UN for research and innovation, especially the Goal 6, focused in Clean 93… “ Provide reference
- The authors acknowledge the reviewer’s comment, and the reference has been added [22].
L142 complex molecules used to colours textiles or products.”
- The authors acknowledge the reviewer’s comment, and the change has been carried out.
L433 Zinc oxide, such as TiO2, is considered…” – Should be corrected.
- The authors acknowledge the reviewer’s comment, and the change has been carried out.
L435: “….ncy in the photo-catalytic degradation of some organic dyes than TiO2…..”- Specify the organic dyes
- The authors acknowledge the reviewer’s comment, and the dyes have been identified, i.e. methyl orange and Rhodamine 6G [193]. Please refer to line 452.
L478: “……..an enhanced heterojunction with another semiconductor for…….” Be specific. Name the narrow band gap semiconducting oxides.
- The authors acknowledge the reviewer’s comment, and the sentence has been changed, and it has been included: “p-type semiconductor, for example CuO”, to… Please refer to line 493.
L591: “Both covalent and non-covalent functionalization processes.” What does it mean?
- The authors acknowledge the reviewer’s comment, and to avoid misleading, the “covalent and non-covalent functionalization processes” terms have been removed and a sentence with some examples of surface functionalization has been added: “Functionalized cellulose has been reported previously with porphyrin, phthalocyanine, polyaniline (PAni), but also embedded with silver nanowires [1,233,234].” Please refer to line 607.
Comment 2: Section 1.3 should be updated with recent work on 1-D semiconducting metal oxide nanostructures and different fabrication techniques as reported in recent articles. Nanofibers is an important 1 D nanostructures. Authors should include some recent references as shown below and highlight the semiconducting metal oxides nanofibers of TiO2, ZnO etc. Authors should include references in the following lines -
- L202-210 : “….several different nanostructures were described….” and in L204: “…..and several other 204 methods have been employed.” Authors need to include articles citing other methods such as gas jet fiber spinning for 1 D nanostructures like nanofibers. Here are some interesting articles for reference:
ChemCatChem 2016, 8, 2525
Materials Science in Semiconductor Processing 42, 1,2016, 98-101
Environ. Sci. Technol. 2015, 49, 3, 1654–1663
Separation and Purification Technology 114, 2013, 108-116
Journal of Environmental Chemical Engineering 9, 2, 2021, 105103
- The authors acknowledge the reviewer’s comment, and the references have been added in the manuscript in section 1.3, but also in 2.1 and 2.2. 1D nanostructures, where nanofibers are included, have been highlighted, with two sentences included. Please refer to lines 214, 301 and 507. Other references associated to metal oxide nanofibers have been included.
Comment 3: L213: TiO2 is the most investigated metal oxide photocatalyst [77], however the use of other 212 semiconductors in photocatalysis, such as ZnO, has been growing lately……” Authors should include relevant references citing ZnO work. Also, authors used “metal oxide” and “semiconductor oxide” in many places throughout the manuscript. It would be better to use one consistent terminology.
- The authors acknowledge the reviewer’s comment, and the references have been added. The term “metal oxide” has been selected and all the manuscript has been altered. Please refer to line 224.
Comment 4: The reviewer thinks that the section 1.1 particularly L221-247 could be precise and short. Some general information about TiO2 structure, crystallinity, band gap etc. should be brief as they are widely available in many other articles. May be authors can highlight some papers for this purpose.
- The authors acknowledge the reviewer’s comment, nevertheless the presentation of TiO2 structure and intrinsic properties are imperative for the further discussions along the manuscript. The relevant papers were cited throughout the 2.1 section.
Comment 5: “….E.g., L275: “……Surface modification, fabrication of composites or 276 heterojunctions also have been reported to expand TiO2 photocatalytic activity under solar…” This statement can be elaborated with relevant examples. Other research work in this topic needs to be cited too. Refer the following recent papers.
Catalysts 2020, 10(2), 227
ChemCatChem 2018, 10, 3305
Applied Surface Science 527, 2020,146780
Chinese Journal of Catalysis 41, 1, 2020, 9-20
Applied Catalysis B: Environmental 259, 2019, 118034
Ceramics International 46, 1, 2020, 38-45
- The authors acknowledge the reviewer’s comment, and the references with photocatalytic processes under visible or solar radiation have been added. Regarding the elaborated statement, some examples have been added, and a sentence indicating some reviews on the topic has been included. Please refer to line 290.
Comment 6: L441-443 and L470-472 have similar message. Authors should double-check it.
- The authors acknowledge the reviewer’s comment, and the second sentence has been changed. Please refer to line 487.
Comment 7: It appears that more than 20 articles by Nunes, D et. al. was used as references in the current manuscript. Although some of these references are relevant in the context of the discussion, the reviewer thinks that the authors could have referred other researcher’s work in the same topic. Cite work done by other researchers in this regard.
- Also, below references do not seem relevant in this review
Ref#75 Nunes, D.; Vilarigues, M.; Correia, J.B.; Carvalho, P.A. Nickel–carbon nanocomposites: Synthesis, structural changes and 827 strengthening mechanisms. Acta Materialia 2012.
Ref#96 Gonçalves, A.; Resende, J.; Marques, A.C.; Pinto, J.V.; Nunes, D.; Marie, A.; Goncalves, R.; Pereira, L.; Martins, R.; Fortunato, E. 872 Smart optically active VO2 nanostructured layers applied in roof-type ceramic tiles for energy efficiency. Solar Energy Materials 873 and Solar Cells 2016, 150, 1-9,
- The authors acknowledge the reviewer´s comment and the references have been removed, and several other have been included throughout the manuscript.
Yours sincerely
Daniela Nunes
Invited Assistant Professor
CENIMAT/i3N, Faculdade de Ciências e Tecnologia - Universidade Nova de Lisboa
CEMOP-UNINOVA
2829-516 Caparica, Portugal
http://www.cenimat.fct.unl.pt/
daniela.gomes@fct.unl.pt

Reviewer 2 Report
Abstract: CO2, TiO2, N2 - 2 should be subscript
line 43 - where this polysaccharide with molecular structure of ((C6H10O5)n is linked together - double parenthesis
line 111 - Could you provide more information about decomposition of inorganic compounds on the semiconductor surface.
line 110 - Why O2• and not O2•−
Fig. 14 e - it is not readeble
There is lack of information about light energy for TiO2 or ZnO - light energy should be equal to or higher than a bandgap of TiO2 or ZnO (380 nm).
Author Response
Dear reviewer,
The authors acknowledge the pertinence of your comments and have revised the paper by taking them into account. The review manuscript has been completely revised and the English has been improved by R. Branquinho, and for that, she has been added as author. All the changes are highlighted. Following the reviewer’s comments order:
Referee 2:
Abstract: CO2, TiO2, N2 - 2 should be subscript
- The authors acknowledge the reviewer’s comment, and the change has been carried out.
line 43 - where this polysaccharide with molecular structure of ((C6H10O5)n is linked together - double parenthesis
- The authors acknowledge the reviewer’s comment, and the change has been carried out.
line 111 - Could you provide more information about decomposition of inorganic compounds on the semiconductor surface.
- The authors acknowledge the reviewer’s comment, and two sentences have been included with organic and inorganic compounds degraded by photocatalysis. Please refer to line 129.
line 110 - Why O2• and not O2•−
- The authors acknowledge the reviewer’s comment, and it has been replaced by O2•−.
Fig. 14 e - it is not readable
- The authors acknowledge the reviewer’s comment, and Figure 14 has been altered.
There is lack of information about light energy for TiO2 or ZnO - light energy should be equal to or higher than a bandgap of TiO2 or ZnO (380 nm).
- The authors acknowledge the reviewer’s comment, and the light source wavelength for both TiO2 and ZnO has been included. Please refer to 108. The band gap of both materials is presented in sections 2.1 and 2.2 for TiO2 and ZnO, respectively.
Yours sincerely
Daniela Nunes
Invited Assistant Professor
CENIMAT/i3N, Faculdade de Ciências e Tecnologia - Universidade Nova de Lisboa
CEMOP-UNINOVA, 2829-516 Caparica, Portugal, http://www.cenimat.fct.unl.pt/ daniela.gomes@fct.unl.pt

Round 2
Reviewer 1 Report
Authors have carefully revised the manuscript to address the reviewer's comment. Therefore, the revised manuscript is recommended for acceptance in Catalysts.
Suggestion:
Some references are missing DOIs -if possible, please add them.